# Estimation System of Disturbance Force and Torque for Underwater Robot Based on Artificial Lateral Line

**Song Kang** [1,*], **Wusheng Chou** [1,2] **and Junhao Yu** [1]

1   School of Mechanical Engineering and Automation, Beihang University, Beijing 100191, China;
    wschou@buaa.edu.cn (W.C.); yujunhao@buaa.edu.cn (J.Y.)
2   The State Key Laboratory of Virtual Reality Technology and Systems, Beihang University,
    Beijing 100191, China
*   Correspondence: kangsong@buaa.edu.cn

**Abstract:** The motion-control precision of a shallow-sea underwater robot is seriously affected by external disturbances such as wind, waves and ocean currents. Due to the lack of a specialized disturbance-sensor system, the disturbance force and torque cannot be sensed effectively. Inspired by bionics, an artificial lateral-line system for estimating external disturbances of an underwater robot is presented in this paper. In the system, the pressure of water is first collected through the pressure-sensor array. Then, the pressure data is processed by a series of algorithms, and the disturbance force and torque are observed from the data. Both multiple linear regression and the artificial neural network method are used to fit the mathematical models of the disturbances. Finally, the system is validated experimentally to be effective and practical. The underwater robot senses the disturbance force and torque from the water indirectly through the artificial lateral-line system, which will improve the accuracy of motion control.

**Keywords:** underwater robot; artificial lateral line; pressure sensor; disturbance estimation

## 1. Introduction

More and more underwater robots are taking the place of human beings in the areas of sea-water fishery, safety search and rescue, scientific investigation and so on [1–3]. The control system of the underwater robot in near shallow seas is hampered by various disturbances—some of which are derived from the outside of the system, such as waves, ocean currents and wind; and some of which are derived from the inside of the system, such as highly nonlinear dynamics and coupling, deviation of model parameters and noise of sensors [4]. Although various disturbance observers and antidisturbance algorithms have been developed by scholars [5,6], it is difficult for underwater robots to obtain disturbance information from inside and outside of the system only by relying on inertial sensors, underwater acoustic sensors and visual sensors. The performance of the underwater robot system is strongly affected by this situation, which makes it difficult to complete fixed-point hovering and track tracking tasks, especially in waters with poor lighting conditions or complicated terrain. From a biological point of view, the lateral-line system is the water-environment sensing organ distributed around the body of fish and other aquatic organisms. The lateral-line system of fish uses a mechanical sensing unit, the nerve colliculus, to perceive the hydrodynamic characteristics around itself and obtain the surrounding environment information through central nervous calculations [7]. Through the bionics principle, the lateral line of aquatic organisms is simulated to perceive the water-environment information, and the lateral-line system of aquatic organisms is transplanted into the underwater-robot system, which will improve the ability of the underwater robot to observe the external-interference information of the system. It is still urgent to solve how to use the existing sensors and computer technology to realize the imitation of the fish lateral-line system and develop the corresponding data-processing methods, so that the

underwater robot can obtain the ability to sense the movement and force of the surrounding water, so as to counter the interference from the external water of the robot.

The first application of underwater robots was in the 1950s. After the Americans developed the "CURV-1" underwater robot, they tried to use underwater robots to salvage hydrogen bombs on the seabed, an extremely dangerous and complex task. After the success of this task, researchers all over the world saw the great national defense research value of underwater robots. Since then, underwater robots have attracted people's attention. They are now widely used in the offshore oil and gas sectors, underwater target search and salvage, underwater patrol inspection and offshore fishery.

The earliest studies of lateral-line systems of fish date back to the 17th century [8]. Hofer first conducted functional research on the lateral line in article [9], and speculated that the lateral line is a receptor to feel the stimulation of water motion. Dijkgraaf initiated the study of the current line-measurement system in 1963 and learned that the lateral line was a kind of independent sensory system, and proposed that the lateral line could feel the vibration stimulation generated by water fluctuation or fluid stimulation generated by the movement of objects [10]. Later researchers have conducted a lot of research on the biological morphology, distribution and function of the lateral-line system, and proposed that the lateral-line system can assist these species in performing various functions such as schooling, prey detection, communication and rheotaxis [11–13].

In recent years, with the rapid development of bionic science, materials and computer science, researchers began to use different kinds of sensors to simulate the lateral-line system of fish to perceive the function of water-flow information [14,15], as is shown in Table 1. The artificial lateral-line system can be used as a scientific bionic experiment to study the interaction mechanism between the biological lateral line and surrounding water flow [16]. It can also be used as a sensor for underwater robots to perceive the environment, and obtain information on the robot's own state and its surrounding water environment. Research of the artificial lateral-line system has attracted more and more attention since 2006, and it is still in its infancy [17]. Sensors based on different kinds of sensing mechanisms, such as piezo-resistive, piezoelectric and optical-detection schemes have been designed by researchers [18–20].

**Table 1.** Previous works about artificial lateral line.

| Author | Achievements |
| --- | --- |
| Abdulsadda et al. [14] | An ALL system based on IPMC sensors was presented to localize the dipole source (vibrating sphere) underwater. |
| Ristroph et al. [16] | The locations with the largest pressure change were concentrated, and a simulation was presented to mimic the flows encountered during swimming. |
| Yang et al. [17] | An ALL system with a distributed linear array was presented, which can successfully perform dipole-source localization and hydrodynamic wake detection. |
| Chen et al. [21] | A dynamic model was developed for the IPMC beam under the flow to estimate fluid properties and flow parameters. |
| Klein et al. [22] | An ALL system equipped with optical-flow sensors was presented to detect water motions, vortices or water (air) movement. |
| Chambers et al. [23] | To detect and navigate turbulent flows and sense fluid interactions for underwater vehicles, a total of 33 pressure sensors were used under Karman vortex streets at different parameters. |
| Venturelli et al. [24] | To sense the underwater environment better, 20 pressure sensors were used to distinguish the Karman vortex streets and uniform flow. |
| Wang et al. [25] | A fish robot with ALL system was presented to detect the beating frequency of its neighboring robot and the distance between the robots. |
| DeVries et al. [26] | Theoretically justified nonlinear estimation strategies were presented to perceive the environment using 4 pressure sensors, and to use the strategies to construct a feedback-control system. |

ALL represents artificial lateral line.

Most of the existing artificial lateral-line systems are composed of multiple water-flow sensors distributed on a cylinder or streamlined shell; the research contents focus on dipole-oscillation source location [21], obstacle identification and tracking [22], flow velocity and Karman vortex-street frequency detection [23], steady and unsteady flow analysis [24] and so on. In the paper [25], the artificial lateral-line system is composed of pressure-sensor arrays distributed on both sides of the body to study the various characteristics of the Karman vortex street. In paper [26], the problem of water-flow perception of the artificial lateral-line system when moving longitudinally and laterally is studied. However, most of the existing work focuses on the research of sensors with different principles and the observation of water-fluid information such as the Karman vortex street, vibration source and water-flow velocity. Research on force and water-flow disturbance of robots has not been studied.

In this paper, the bionic-sensing theory and method of water environment based on the artificial lateral system is used to design an artificial lateral system with a reasonable bionic mechanism and full fluid-sensing function for underwater robots. Moreover, by means of theoretical analysis, mathematical modeling and experiments, the data collected by the artificial lateral-line system are analyzed to perceive the interaction between the underwater robot and the water body through the line-measuring system. Meanwhile, an experimental platform for the artificial lateral system is built. In addition, different experimental conditions are set for sway, surge and yaw movements. Furthermore, experimental data are preprocessed, and models based on the linear regression method and artificial neural network method can be obtained respectively on the basis of physical models and experimental data. Finally, the accuracy, advantages and disadvantages of these two models are briefly explained. The main content of this paper and the sequence of ideas are shown in Figure 1.

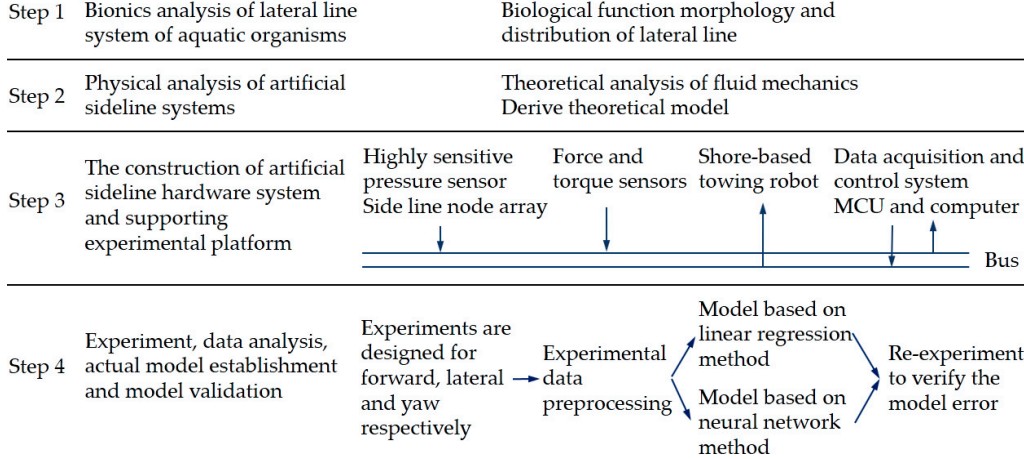

**Figure 1.** Main content and order of this paper.

The rest of this paper is organized as follows: The biological mechanism of the artificial lateral system and the design of the hardware of the artificial lateral system are briefly described in Section 2. In Section 3, the experimental platform and the setting of the experiment are given. Experimental data and their preprocessing are presented in Section 4. Experimental data-processing methods based on linear regression and artificial neural network methods are carried out respectively in Sections 5 and 6. Finally, conclusions are drawn in Section 7.

## 2. Lateral-Line System and Artificial Lateral-Line System

### 2.1. Lateral Lines of Aquatic Organisms

Fish and aquatic amphibians live in underground rivers, in dark lakes, on the seafloor and in murky waters where light is absent. In this environment, the visual perception

system is greatly limited. However, in order to better complete the complex behaviors of feeding, avoiding enemies, reproduction, clustering, navigation and communication, some fish and aquatic amphibians have evolved a special sensory system [27], namely the lateral line.

From the micrograph, the basic unit of the sensory organ of the lateral-line system is the neuromast, which is composed of hair cells, supporting cells and mantle cells, as shown in Figure 2b. Hair cells have cilia on their tips, and the cilia are coated with a gel secreted by Sertoli cells and mantle cells and from the cupula structure together. Ciliated fascicles are the receptor units of the thalamus cells. When fluid flows over the surface of the fish, it causes the cilia bundles to deflect or vibrate. Hair cells convert mechanical stimulation into electrophysiological signals and transmit them to the central nervous system.

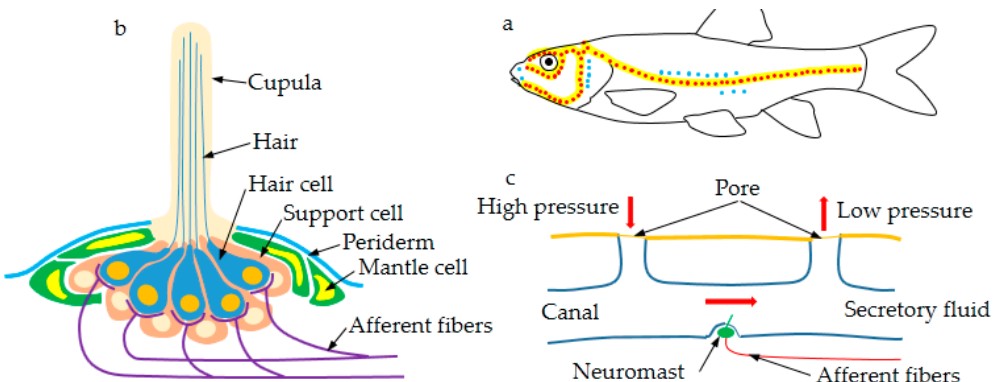

**Figure 2.** Brief introduction of a fish's lateral line [28]. (**a**) Distributions of superficial neuromasts (blue dots) and canal neuromasts (red dots); (**b**) The microstructures of an individual neuromast; (**c**) A canal neuromast located in a fluid-filled lateral-line canal.

In terms of function, there are two types of colliculus, namely superficial neuromasts and canal neuromasts. The ductal-nerve colliculus is generally distributed in the lateral canal below the surface of the fish's body. The lateral-line canal is filled with liquid and connected to the water through the lateral-line pore. When the pressure is different between the two sides of the lateral-line pore, it causes the fluid inside the lateral-line canal to flow, which causes the hair cells to swing and triggers the nerves in the neuromast to respond, as shown in Figure 2c. It is generally believed that the ducted-nerve colliculus is sensitive to high-frequency information, such as the acceleration of fluid, by sensing the pressure difference between two adjacent lateral pores to sense the flow information of fluid parallel to the body axis. Therefore, the canal neuromast senses the flow information of fluid parallel to the body axis by sensing the pressure difference between two adjacent lateral pores. The canal neuromast is sensitive to high-frequency information, such as fluid acceleration.

Lateral neuromasts are numerous and spread throughout the fish's surface in a specific pattern, as is shown in Figure 2a. Typically, there are seven lateral-line branches in the head, and two pairs of lateral line branches in the trunk and caudal. Superficial neuromasts are distributed around the canal neuromasts, and the arrangement of superficial neuromasts and canal neuromasts is at an angle of about 90°. The special structure and distribution characteristics of the lateral lines of fish help them to learn the temporal and spatial changes of surrounding water and collect hydrodynamic information in an all-round way.

### 2.2. Physical Device of Artificial Lateral-Line System Suitable for the Underwater Robot

The working environment of underwater robots is complex. Underwater robots must be equipped with sophisticated environmental-awareness systems, to have a good sense of adaptation to the underwater environment and perform underwater operational tasks. Due to the particularity of the water environment, many sensors are difficult to work effectively under water, which restricts the development of the water-environment sensing technology of the underwater robot. At present, underwater robots mainly use

two kinds of perception technology: underwater acoustic perception and visual perception. Underwater acoustic sensing technology, also known as sonar technology, is a widely used underwater sensing system. However, due to its large volume; high energy consumption; and ease at being affected by propagation attenuation, multipath effects, reverberation interference and ocean noise, a sonar transducer is difficult to use in small underwater robots. Visual perception is the direct acquisition of images by underwater cameras with rich and intuitive information, which has been applied to some small underwater robots. However, due to the influence of weak light and the complex environment underwater, the application is limited. The environment of underwater vehicles and aquatic creatures is very similar. Traditional underwater robots lack systems to sense the water around them, while the artificial lateral-line system can make up for such defects. In order to perceive environmental information of the surrounding water, we use pressure sensors to develop an artificial lateral-line system based on the fish lateral-line system to simulate the function of the fish lateral-line system [29]. The artificial lateral-line system will be an effective sensing system which cannot be replaced by traditional sensors.

### 2.2.1. Introduction of Underwater Robot System

The underwater robot used in this paper is shown in Figure 3. The Myring curve equation is used as the contour curve of the underwater robot. The whole is divided into three parts: the entrance section, middle section and run section, with a total length of 1.8 m. Six thrusters surround the body of the underwater robot in three pairs, two in each pair arranged in parallel. Among them, thrusters 1 and 2 provide the main thrust force for the robot's surge. Thrusters 3 and 4 are mainly for the robot's sway and yaw movement. Thrusters 5 and 6 are mainly for the heave movement.

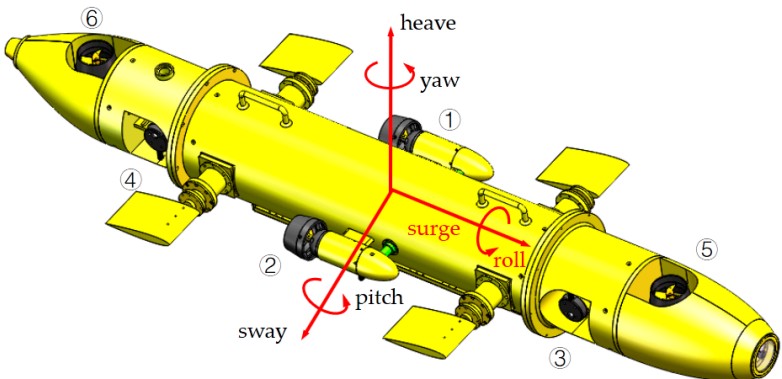

**Figure 3.** Sketch of the underwater robot. The six thrusters are labeled as 1–6.

The underwater robot also includes four fins to assist in underwater propulsion and control. Because the thrust generated by the fins is relatively small compared with the propeller, it will not be considered in this paper.

Similar underwater vehicles also appeared in [30,31]. These papers discussed the problems of depth-determination control and antidisturbance control of the underwater vehicle.

### 2.2.2. Introduction of the Artificial Lateral-Line System of Underwater Robot

In this paper we mainly focus on the estimation of the interference force and torque of the water environment to the robot by using the artificial lateral-line system. Specifically, the pressure-sensor array is used to measure the pressure at different positions of the underwater vehicle. The hydraulic information is then deduced and the external disturbance force of the robot is estimated, according to the value and position of the pressure at each point and the relationship between the pressure at each point.

A micro-pressure sensor is chosen to collect the pressure at the key points of the robot body. Moreover, a total of 9 pressure sensors are installed on both sides and the nose tip of the robot to form a pressure-sensor array, in order to establish an artificial lateral-line

system on the robot's body. Specifically, the model of the commercial pressure sensor is ms5803-01ba from TE Connectivity Ltd. The resolution of the pressure sensor is 1.2 PA and it is sensitive to tiny hydrodynamic pressure variations in water.

The nine pressure sensors are labeled as $P_0$, $P_{L1}$, $P_{L2}$, $P_{L3}$, $P_{L4}$, $P_{R1}$, $P_{R2}$, $P_{R3}$ and $P_{R4}$. They represent the node of the nose, the four nodes on the left and the four nodes on the right. The position of the pressure sensors at the nose tip and both sides of the robot is shown in Figure 4. The pressure data are read to the MCU through the IIC bus, and then sent to the computer onshore for further storage and analysis through the wired RS485 serial port.

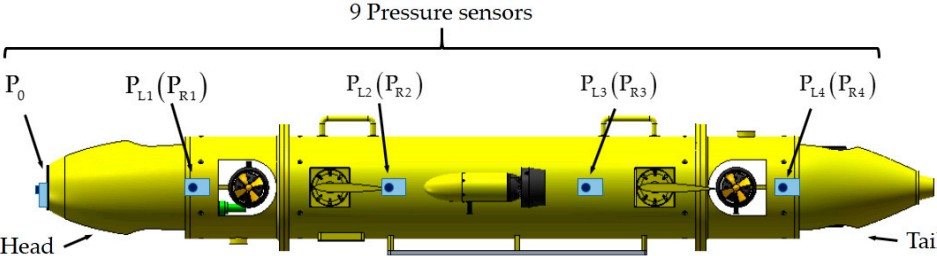

**Figure 4.** The position of pressure sensors. The nine pressure sensors are labeled as $P_0$, $P_{L1}$, $P_{L2}$, $P_{L3}$, $P_{L4}$, $P_{R1}$, $P_{R2}$, $P_{R3}$ and $P_{R4}$.

### 2.3. Experimental Platform for Artificial Lateral-Line System of Underwater Robot

In order to find out the relationship between data which are collected by the artificial lateral-line system and the external disturbance force and torque of the robot, these data need to be measured simultaneously. Moreover, stable relative motion between the underwater robot and the water is necessary. Based on the robot and the artificial lateral-line system, a set of traction-force-measuring experiment platforms is constructed. It consists of a mobile traction platform and a force-and-torque-measuring system. The experiment platform is shown in Figure 5.

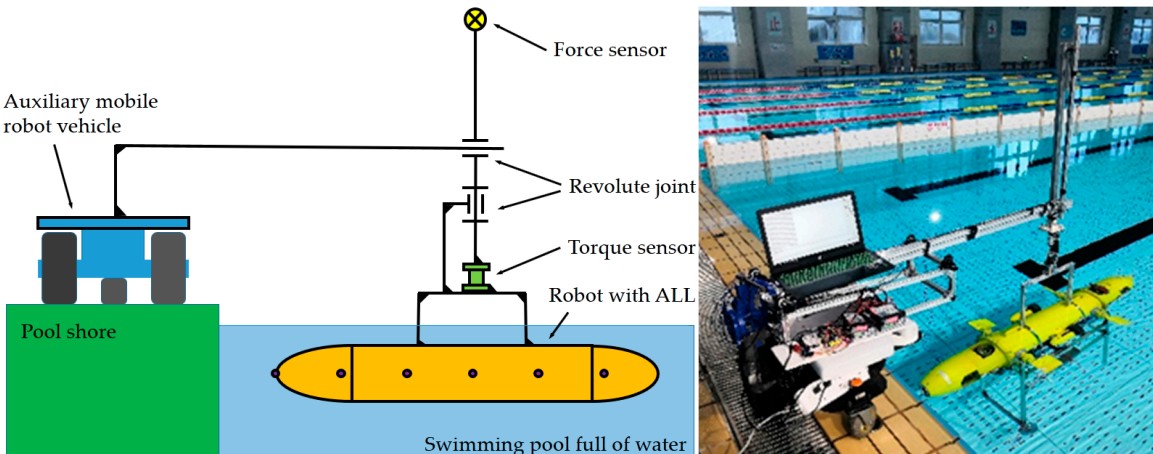

**Figure 5.** Experimental platform for artificial lateral-line system.

The mobile traction platform is mainly composed of a mobile robot onshore. The mobile robot drives along the bank of the experimental pool at a set speed. It is connected to the underwater robot through a set of connections. The towed underwater robot moves synchronously with the mobile traction platform, and the underwater robot moves in a straight line at a certain speed. The force-and-torque-measuring system is a mechanism with two rotating-motion pairs. The forces and torques applied to the underwater robot are simultaneously measured by a force sensor and a torque sensor. Force and torque are decoupled by rotating-motion pairs and linkages so that they will not interfere with each other. The force is transmitted to the force sensor at the top through an equal-arm lever.

Data of the artificial lateral-line system and force and torque sensors are recorded simultaneously when the mobile traction platform and the underwater robot move together.

## 3. Methods of Artificial Lateral-Line System for the Underwater Robot

In the previous section, the data of the artificial lateral-line system and the sensors of force and torque are obtained. However, obtaining a connection between these data requires further study.

The center of buoyancy of the underwater robot we use is much higher than the center of gravity, and its motion in pitch-and-roll directions has better anticapsule capability. It can be considered stationary and self-stabilizing in these two degrees of freedom [32]. Therefore, the degrees of freedom of the body of the underwater robot are reduced from six to four, namely the yaw, surge, heave and sway motions. Since the main motion of the wave and current is concentrated in the horizontal plane, only the interference and influence on the horizontal plane of the underwater robot are considered. In other words, the influence of water on the robot's heave motion is ignored. Disturbance forces and torques in the longitudinal, transverse and yaw directions are discussed, respectively.

### 3.1. Relationship between Disturbance Force and Lateral-Line-System Data on Surge

The cross-sectional area is relatively small in the direction of surge. The force caused by water can be approximated as

$$F_{surge} = K(p - p_a)A \tag{1}$$

where $p$ represents the pressure measured by the pressure sensor node of the artificial lateral-line system set on the nose of the robot; $p_a$ is hydrostatic pressure which can be obtained from the lateral-line system; $A$ represents the cross-sectional area in the direction of surge; and $K$ represents the parameters that need to be verified by experiments.

From Equation (1) we can come to the conclusion that the only constant to be calibrated is $K$ in this equation, and the relationship between $F_{surge}$ and $p - p_a$ is linear.

### 3.2. Relationship between Disturbance Force and Lateral-Line-System Data on Sway

The sensitivity of the robot to water disturbance varies greatly in different directions as the difference in cross-sectional area. In the sway direction, the cross-sectional area of the robot is much more than in the surge direction.

In this paper, water is supposed to be an ideal fluid. The analysis of disturbance force in this direction can be approximated as flow around circular cylinders, as the main robot body is cylindrical.

The expression of flow resistance around circular cylinders is

$$F_{sway} = A_D C_D \frac{r v_0^2}{2} \tag{2}$$

where $A_D$ represents the cross-sectional area of the robot in the sway direction; $C_D$ is the coefficient of flow around circular-cylinder resistance; $\rho$ represents the fluid density; and $v_0$ is the fluid velocity [33].

For a point in an ideal fluid from Bernoulli's equation [34]:

$$p + \frac{1}{2}\rho v^2 + \rho g h = C \tag{3}$$

where $h$ represents the depth of water, and $C$ is a constant.

For the artificial lateral-line-system node on the facing-flow side, its fluid velocity is zero. Thus, the pressure at this node is

$$p = p_a + \frac{1}{2}v_0^2 \tag{4}$$

Substitute equation A into equation B and obtain

$$F_{sway} = A_D C_D (p - p_a) \tag{5}$$

where $A_D$ is the cross-sectional area of the robot in sway direction; $p$ and $p_a$ can be obtained from the artificial lateral-line system; and $C_D$ represents the only parameters that need to be verified by experiments in this equation. Additionally, the relationship between $F_{sway}$ and $P - Pa$ is linear.

### 3.3. Relationship between Disturbance Torque and Lateral-Line-System Data on Yaw

The force of the underwater robot in the sway direction is not uniform, as it takes the shape of a strip and the wave and current are not uniform. When this happens, disturbance torque is generated in the yaw direction, as shown in Figure 6. In order to obtain the disturbance torque in the yaw direction, the inhomogeneous part of a disturbance force is integrated in the surge direction.

$$M_{yaw} = \int_0^L \Delta F dl \tag{6}$$

where $\Delta F$ is the inhomogeneous part of the disturbance force in the surge direction. $l$ is the length of the underwater robot.



**Figure 6.** The equivalent relationship between uneven force, uniform force with torque and torque-center-position changes.

Based on the discussion and theoretical analysis above, the relationship between the forces and torques of the underwater robot and the underwater-pressure data collected from the artificial lateral-line system for the underwater robot are deduced in surge, sway and yaw directions. However, some coefficients in these equations need to be obtained by further experiments.

## 4. Experiment and Data Preprocessing of Artificial Lateral-Line System

### 4.1. Experiment of Artificial Lateral-Line System for the Underwater Robot

In the previous section, the relationship between the artificial lateral-line system and sensors of force and torque is roughly obtained. Meanwhile, coefficients in those equivalents need to be calibrated by further experiments. Therefore, three sets of experiments are conducted. The material in all sets of experiments is the same, but the device is set upon different sets of experiments.

The first set of experiments is regarding the relationship between the force in direction of surge and the data of the artificial lateral-line system. The experiment is set up as follows: the bow direction of the underwater robot is consistent with the movement direction of the underwater robot, and the axis of the force-measuring device passes through the center of the underwater robot, as shown in Figure 7a. The external force and torque on the underwater robot are mainly force; the torque is zero. The direction is opposite to the direction of motion.

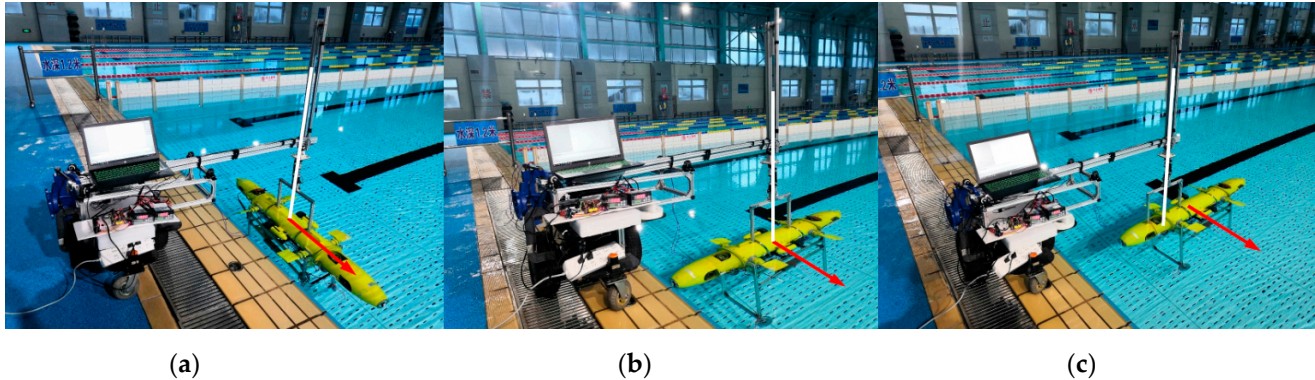

**Figure 7.** Three sets of experiments. The red arrow in the figure is the movement direction of the underwater robot and the white line is the axis of the force-and-torque-measuring device. (**a**–**c**) show the experiments in directions of surge, sway and yaw respectively.

The experiment regarding the force in the direction of sway and the data of the artificial lateral-line system is conducted secondly. The bow direction of the underwater robot and the movement direction of the underwater robot are perpendicular. As with the previous experiment, the axis of the force-measuring device passes through the center of the underwater robot, as shown in Figure 7b. In this experiment, force is dominant, and torque is zero in theory.

In the third set of experiments, the relationship between the torque in the direction of yaw and the data of the artificial lateral-line system is studied. The bow direction of the underwater robot is perpendicular with the movement direction. As shown in Figure 7c, the axis of the force-measuring device does not pass through the center of the underwater robot, which is kept at a distance. In this set of experiments, torque is the main object to be studied.

*4.2. Experimental Results and Data Preprocessing*

In the section above, three sets of experiments are conducted to obtain the data of the artificial lateral-line system and the data from sensors of force and torque. Although temperature compensation and other anti-interference measures are taken, the noise from sensors is still unavoidable. The data of the pressure sensor, torque sensor and force sensor are constantly fluctuating and not stable. Moreover, abnormal data points may occasionally occur. Experimental data need to be filtered before data fitting.

Useful signals in this experiment are concentrated at low frequencies, which can be drawn by theoretical analysis and observation of collected data. Therefore, a low-pass filter is designed for information preprocessing. The Butterworth filter is adopted, and an analog filter is converted to the digital filter by the bilinear transformation method [35].

As the most important sensors of the artificial lateral-line system, the pressure-sensor data-sampling frequency is $f_s = 15$ Hz. The passband cut-off frequency is set to $f_c = 1.5$ Hz. The stopband start frequency is set to $f_z = 6.75$ Hz. Thus, the digital angular frequency can be calculated as

$$\begin{cases} \omega_c = \frac{2\pi f_c}{f_s} = 0.2\pi \\ \omega_z = \frac{2\pi f_z}{f_s} = 0.9\pi \end{cases} \tag{7}$$

Distortion occurs on the digital frequency axis of the frequency response curve of the filter, when the s-plane imaginary axis is converted to the Z-plane unit circle using a nonlinear transformation. In order to solve this problem, the simulated angular frequency is predistorted, and the predistorted simulated angular frequency is as follows:

$$\begin{cases} \Omega_c = \frac{2}{T_s} \tan \frac{\omega_c}{2} \\ \Omega_z = \frac{2}{T_s} \tan \frac{\omega_z}{2} \end{cases} \tag{8}$$

Passband signal attenuation is 3 dB, stopband attenuation is 15 dB. Then, the final digital Butterworth filter is obtained:

$$\frac{Y(z)}{X(z)} = H(z) = \frac{0.5329z + 0.5329}{z + 0.0658} \tag{9}$$

Convert to difference-equation form:

$$y(n) = k_1 y(n-1) + k_2 x(n) + k_3 x(n-1) \tag{10}$$

where $k_1 = -0.0658$, $k_2 = 0.5329$ and $k_3 = 0.5329$.

As shown in Figure 8, the amplitude remains unchanged from 0 to 2.25 HZ, and then decreases. Under this condition, high-frequency noise will be filtered out when the frequency of load change is not large and the phase-frequency curve is close to linear.

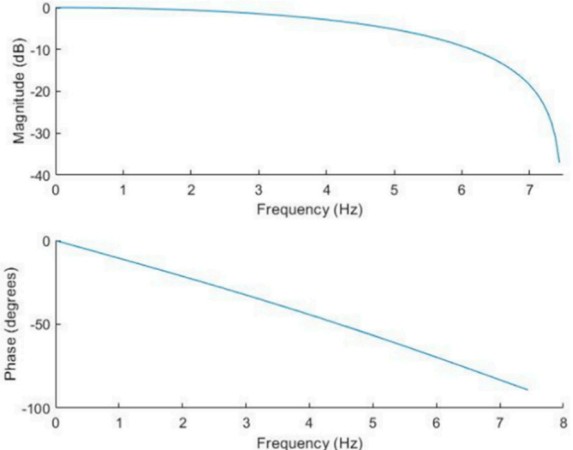

**Figure 8.** Frequency response curve of digital filter.

The filter that is designed above meets the requirements of low-distortion transmission, the amplitude frequency of the signal is basically unchanged and the phase frequency is linear.

## 5. Experimental Data Analysis Based on Linear Regression

### 5.1. Data Analysis of Disturbance Force and Lateral-Line System Data on Surge

As mentioned in Section 4.1, the first set of experiments is about the relationship between disturbance force and the data of the artificial-line system in surge. The underwater robot used in this experiment is cylindrical with a large length-to-diameter ratio, so that the torque in yaw caused by the uneven force in the surge direction can be ignored.

The data of sensor $p_0$ located at the bow surface of the ROV is for the pressure data of the facing-flow surface. The mean value of $p_{L1}$, $p_{L2}$, $p_{L3}$, $p_{L4}$, $p_{R1}$, $p_{R2}$, $p_{R3}$ and $p_{R4}$ is taken as the pressure on the non-facing-flow surface.

$$p_{surge} = p_f - p_{nf} \tag{11}$$

where $p_f = p_0$, and $p_{nf}$ is the mean value of $P_{L1}$, $P_{L2}$, $P_{L3}$, $P_{L4}$, $P_{R1}$, $P_{R2}$, $P_{R3}$ and $P_{R4}$.

The relationship between $p_{surge}$ and disturbance force $F_{surge}$ is analyzed by the multiple linear regression method, and the regression equation is derived as (12).

$$F_{surge} = a_1 p_{surge} - b_1 \tag{12}$$

where $a_1$ and $b_1$ are the parameters obtained by multiple linear regression [36].

### 5.2. Data Analysis of Disturbance Force and Lateral-Line-System Data on Sway

The second set of experiments is to calibrate parameters in Section 3.2, which determines the relationship between the lateral disturbance force and the data of the measuring line system.

In this group of experiments, the lateral face of the robot is the facing-flow surface, and the bow surface is the non-facing-flow surface.

$$p'_i = p_i - p_a \tag{13}$$

where $p_a$ is the value of the artificial lateral-line node $p_0$ which is located at the nose tip of the robot. $p_i$ is the data of $p_{L1}$, $p_{L2}$, $p_{L3}$, $p_{L4}$, $p_{R1}$, $p_{R2}$, $p_{R3}$ and $p_{R4}$.

The linear relationship between the force of the robot in sway and $p_i$ is proved in Section 3.2. The profile of an underwater vehicle in a lateral direction is a plane surrounded by an Myring curve. The artificial lateral-line system can only obtain pressure data at key points of the surface. That is not a representation of the pressure on the whole surface directly. Therefore, these points need to be interpolated to obtain a set of relatively complete data $p'$. After interpolation, all the data are integrated with the area of the side section to obtain the required disturbance parameters $A_{left}$ and $A_{right}$.

$$A = \int_0^L r(l)p'dl \tag{14}$$

where $r(l)$ is the Myring profile curve equation of the underwater robot. $L$ is for the length of the robot.

Multiple linear regression is performed using the measured disturbance data $F_{sway}$ and the disturbance parameter $A_{left}$ and $A_{right}$. The regression equation is obtained [37], and $a_2$, $a'_2$ and $b_2$ are the parameters obtained by multiple linear regression.

$$F_{sway} = a_2 A_{left} - a'_2 A_{right} - b_2 \tag{15}$$

### 5.3. Data Analysis of Disturbance Force and Lateral-Line-System Data on Yaw

The third group of experiments is to find out the relationship between the disturbance torque in the yaw direction and the data of the artificial lateral-line system. In this set of experiments, the rotation axis in yaw of the robot and the force-measuring device are not coincided with the geometric center of the underwater robot. It is used to simulate the yaw torque of an underwater robot when it is subjected to an unbalanced lateral force. After interpolating the data of the artificial lateral-line system, $p'$ is obtained as in the yaw direction. The left and right disturbance parameters of torque are defined as $B_{left}$ and $B_{right}$; furthermore,

$$B = \int_0^L r(l)p'\Delta l dl \tag{16}$$

$\Delta l$ is the distance from the point being integrated to the yaw axis.

Multiple linear regression is performed using the parameters $B_{left}$, $B_{right}$ and the torque $M_{yaw}$ obtained from the torque-measuring platform, and the regression equation is derived, as is shown in Equation (16).

$$M_{yaw} = a_3 B_{left} - a'_3 B_{right} - b_3 \tag{17}$$

### 5.4. Error Analysis of Linear Regression

After the training data have been trained using linear regression, the test data in other experiments are obtained to verify the predicting error of this method. Predicting scatter images are shown in Figure 9.

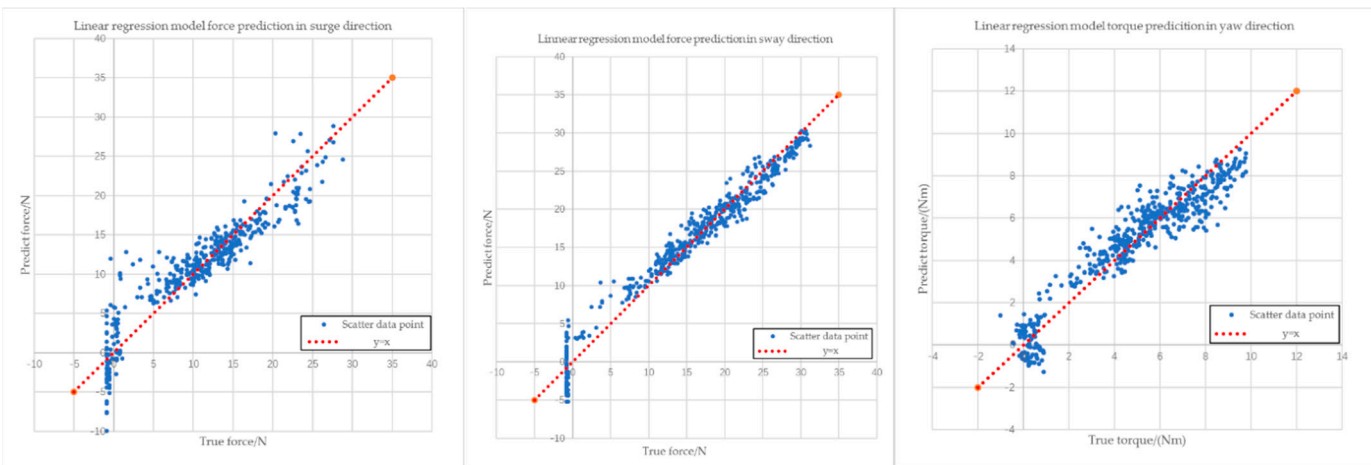

**Figure 9.** Regression models predicting scatter images in surge, sway and yaw directions.

The horizontal axis represents the real disturbance value of the robot measured by the force/torque sensor; the vertical axis represents the disturbance value predicted by the model; the red line is the y = x line; each blue point represents a set of artificial lateral-line data; its abscissa is the corresponding real value of the disturbance; and the ordinate is the predicted value of the disturbance output by the model. Theoretically, the closer the blue point is to the y = x line, the closer the disturbance value predicted by the model is to the real disturbance value, and the better the prediction effect of the model is. On the contrary, the farther the blue point is from the y = x line, the more the disturbance value predicted by the model deviates from the real disturbance value, and the worse the prediction effect of the model is. The distance from the blue point to the red line in the *Y*-axis direction directly reflects the model error.

Predicting-error analysis of the regression models obtained by this method is shown in Table 2.

**Table 2.** Predicting error of the regression models.

| Direction | SSE | MSE | MAE |
|---|---|---|---|
| surge | 3160.00 $N^2$ | 7.09 $N^2$ | 1.96 N |
| sway | 2359.73 $N^2$ | 4.42 $N^2$ | 1.45 N |
| yaw | 417.72 $(Nm)^2$ | 0.81 $(Nm)^2$ | 0.73 Nm |

SSE is the sum of squares due to error, namely the sum of the squares of the differences between the dependent variable and the regression values of the dependent variable; MSE is the mean squared error which reflects the accuracy of regression estimation; MAE is the mean absolute error, the average value of the absolute error between the observed value and the real value.

## 6. Experimental Data Analysis Based on Artificial Neural Network

The multiple linear regression model is used to analyze the relationship between the data of the artificial lateral-line system and the data of force and torque sensors on the robot, in the previous content of this paper. However, the premise of using this method is to prove that the system is linear. In the process of proving that the system is linear, we approximated the robot body as a cylinder and water as an ideal fluid. In practice, the shape of the robot is not an absolute cylindrical shape, and nonlinear terms are introduced in the system by its hydrofoils and cables. Nonlinear errors in these systems cannot be eliminated by training the model by using multiple linear regression methods. The advantage of neural networks is that no theoretical analysis is required to confirm the relationship between input and output, because both linear and nonlinear systems can be learned. When the system is

nonlinear, a higher accuracy will be achieved by using the artificial neural network [38]. Therefore, the method of the BP neural network is used to further analyze.

A three-layer neural network is created according to the actual demand. The first layer is the input layer, which receives the data of the artificial line system, as is shown in Figure 10. 128 neurons and 64 neurons are in the second and the third layer, respectively, and the last layer is the output layer with 1 neuron. A sigmoid function is selected as the activation function. Furthermore, the mean square error is selected as the loss function. Three models are trained according to different inputs and outputs, which correspond to the three motion components of the degree of freedom.

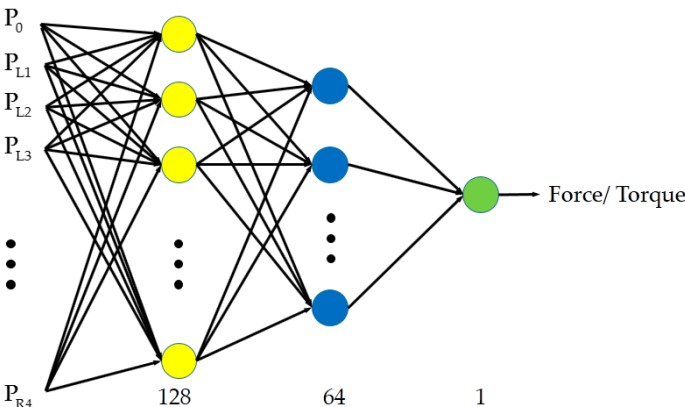

**Figure 10.** Structure of artificial neural network used.

After the model has been trained in training data using the artificial neural network, the predicting errors of the model are analyzed in the test-experiment data. The predicting scatter images in surge, sway and yaw directions are shown in Figure 11. The predicting-error analysis of the model in test data by using the artificial neural network is shown in Table 3.

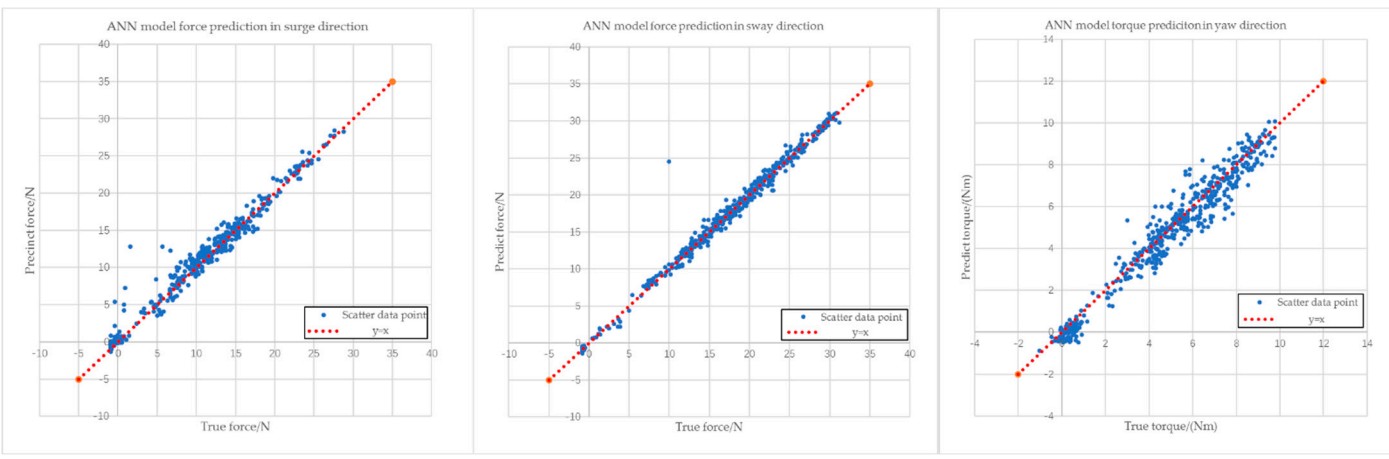

**Figure 11.** The ANN models predicting scatter images in surge, sway and yaw directions.

**Table 3.** Predicting error of artificial neural network.

| Direction | SSE | MSE | MAE |
|-----------|-----|-----|-----|
| surge | 674.41 $N^2$ | 1.51 $N^2$ | 0.77 N |
| sway | 412.34 $N^2$ | 0.77 $N^2$ | 0.48 N |
| yaw | 200.95 $(Nm)^2$ | 0.39 $(Nm)^2$ | 0.47 Nm |

### 7. Conclusions

As we can summarize from the model and error analysis of the system, the system accuracy of the multiple linear regression method is lower, since the basis of using the multiple linear regression model to process experimental data is that the system is linear. However, some uncertain nonlinear factors in the actual experimental process result in inaccurate models and larger errors. Nevertheless, some advantages of the multiple linear regression method must be acknowledged; a smaller scale of calculation and higher real-time performance must be mentioned, and low-computing-power equipment can be used to build platforms such as single-chip microcomputers. Moreover, the system can be directly assumed as a black box without the tedious theoretical derivation. At the same time, with a higher system accuracy, the drawbacks of the artificial neural network method are also obvious. Larger scales of calculation and an equipment platform with higher computing power are required. This shortcoming is particularly prominent in the underwater robot with narrow internal space.

In this paper, to estimate the disturbance of force and torque from the wave and current, an artificial lateral-line system is constructed by using multi-sensor pressure nodes, which are inspired by the lateral-line system in fish. The model of underwater vehicle equipped with an artificial lateral-line system is theoretically analyzed from the perspective of mechanics, and an experimental platform for the artificial lateral system is built. Different experiments are conducted for the sway, surge, and yaw motion components. Experimental data are preprocessed by the Butterworth filter, and models based on the linear regression method and the artificial neural network can be obtained respectively on the basis of physical models and experimental data. The accuracy, advantages and disadvantages of these two models are briefly explained at the end.

The work in this paper provides a theoretical basis and implementation method for underwater robots to obtain a perception system of the water environment, which can be used in antidisturbance control of underwater robots. The stability of underwater robots will be improved through that method, which will make a difference for underwater tasks such as underwater target recognition and underwater manipulator operation.

In following work, the artificial lateral-line system will be further improved. Combined with the artificial lateral-line system and disturbance observer in the control algorithm method, the external and internal disturbances of the underwater robot will be observed at the same time, which will strengthen the antidisturbance ability of the underwater robot.

**Author Contributions:** Conceptualization, W.C. and S.K.; methodology, S.K.; software, J.Y.; validation, S.K., W.C. and J.Y.; formal analysis, S.K.; investigation, S.K.; resources, S.K.; data curation, J.Y.; writing—original draft preparation, S.K.; writing—review and editing, W.C.; visualization, S.K.; supervision, W.C.; project administration, W.C.; funding acquisition, W.C. All authors have read and agreed to the published version of the manuscript.

**Funding:** This research was funded by the National Natural Science Foundation of China under grant number [61633002].

**Institutional Review Board Statement:** Not applicable.

**Informed Consent Statement:** Not applicable.

**Data Availability Statement:** Not applicable.

**Acknowledgments:** This work was supported by the National Natural Science Foundation of China under grant number [61633002].

**Conflicts of Interest:** The authors declare no conflict of interest.

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
