# Peer review of "Estimation System of Disturbance Force and Torque for Underwater Robot Based on Artificial Lateral Line"

_applsci, doi:10.3390/app12063060_

Round 1

Reviewer 1 Report

Comments:

The topic is interesting. “Estimation System of Disturbance Force and Torque for Under-water Robot Based on Artificial Lateral Line”. The authors aimed to overcome the lack of a specialized disturbance sensor system, the disturbance force and torque cannot be sensed effectively. So, the authors used an artificial lateral line system for estimating external disturbances. That is interesting and for improving the quality of the paper, the authors can address the following comments:

  1. In the abstract, the reviewer would like to read about the significance of the results or findings? This can be one or two statements at the end of the abstract.
  2. In keywords, there are many keywords and I suggest the author can skip multiple linear regression; “artificial neural network” because authors already used the keyword “Artificial Lateral Line”
  3. In the introduction lines 26, 27, the authors stated a few external and internal effects the underwater robot and used etc. I would like to see more external and internal effects highlighted not “etc”. This is to clearly review the problem statement.
  4. The author might include a table in the introduction include a previous work. The presentation of the literature review in the paper is a little boring.
  5. The author/s can highlight the applications of the underwater robot in the introduction to show the significance of this research contribution.
  6. The two paragraphs of lines (85-97) and (100-106) can combine together in one paragraph. I like the way of summarising the paper in paragraph lines (85-97) but need to include which section as presented in paragraph lines (100-106)
  7. The discussion and conclusion can be separated. I can not see a discussion in section7 but only a conclusion and can be one paragraph.
  8. The manuscript is technical, and the paper presentation can be improved.

Author Response

Response to Reviewer 1 Comments

[Song Kang

Beihang University,

XueYuan Road No.37,

HaiDian District, BeiJing, China

kangsong@buaa.edu.cn]

Re: Response for manuscript applsci-1628455 “Estimation system of disturbance force and torque for underwater robot based on artificial lateral line” by Kang Song, Wusheng Chou, and Junhao Yu

Dear Reviewers :

Thank you very much for your time involved in reviewing the manuscript and your very encouraging comments on the merits.

Comments:

The topic is interesting. “Estimation System of Disturbance Force and Torque for Under-water Robot Based on Artificial Lateral Line”. The authors aimed to overcome the lack of a specialized disturbance sensor system, the disturbance force and torque cannot be sensed effectively. So, the authors used an artificial lateral line system for estimating external disturbances.

We also appreciate your clear and detailed feedback and hope that the explanation has fully addressed all of your concerns. In the remainder of this letter, we discuss each of your comments individually along with our corresponding responses.

To facilitate this discussion, we first retype your comments in italic font and then present our responses to the comments.

Comment 1:

In the abstract, the reviewer would like to read about the significance of the results or findings? This can be one or two statements at the end of the abstract.

Response 1:

Thanks for your great suggestion on improving the accessibility of our manuscript. We have added some statements at the end of the abstract. The relevant contents are provided below as a screen dump for your quick reference.

Comment 2:

In keywords, there are many keywords and I suggest the author can skip multiple linear regression; “artificial neural network” because authors already used the keyword “Artificial Lateral Line”

Response 2:

We think what you have suggested about the keywords is a very sound one. So that we have removed the key words: “multiple linear regression” and “artificial neural network”.

Comment 3:

In the introduction lines 26, 27, the authors stated a few external and internal effects the underwater robot and used etc. I would like to see more external and internal effects highlighted not “etc”. This is to clearly review the problem statement.

Response 3:

According to the information consulted, we have removed the “etc”, and added a “noise of sensors” in ”disturbances of which are derived from the inside of the system”.

Comment 4:

The author might include a table in the introduction include a previous work. The presentation of the literature review in the paper is a little boring.

Response 4:

Thanks again for your great suggestion on improving the accessibility of our manuscript. We have added atable about the previous works. And the relevant contents are provided below as a screen dump for your quick reference.

Comment 5:

The author/s can highlight the applications of the underwater robot in the introduction to show the significance of this research contribution.

Response 5:

We have added atable about the applications of the underwater robot in the introduction. And the relevant contents are provided below as a screen dump for your quick reference.

Comment 6:

The two paragraphs of lines (85-97) and (100-106) can combine together in one paragraph. I like the way of summarising the paper in paragraph lines (85-97) but need to include which section as presented in paragraph lines (100-106)

Response 6:

We have combine the two paragraphs together in one paragraph, and rewrite it in the way of paragraph lines (85-97).

Comment 7:

The discussion and conclusion can be separated. I can not see a discussion in section7 but only a conclusion and can be one paragraph.

Response 7:

Your suggestion is reasonable. There is no discussion in section 7. So we changed the title of Chapter 7 and added some discussion to the previous part.

Comment 8:

The manuscript is technical, and the paper presentation can be improved.

Response 8:

We have checked this manuscript, and many statements have been changed. We think the presentation of this paper have improved somehow.

We would like to take this opportunity to thank you for all your time involved and this great opportunity for us to improve the manuscript. We hope you will find this revised version satisfactory.

Sincerely,

Song Kang, Wusheng Chou, and Junhao Yu

Reviewer 2 Report

  1. This paper presents an artificial lateral line system to estimate external disturbances of an underwater robot, where multiple linear regression and artificial neural network method are used to fit the mathematical models of the disturbances.
  2. This study is interesting, and the authors use experimental data to support their results. Some minor things might be improved, which will enhance the qualities of this paper:
  3. The format and types of equations are not like those expressed in most literature. For instance, variables should be expressed in italic symbols, and symbol * is not usually used to express multiplication unless its refers to other specific mathematical operations. Or, maybe this journal defines a specific format and type for equations, and the authors have followed the rules.
  4. It looks like that all equations provided are very fundamental, so this paper looks like that there are many theoretical theorems developed and provided. I would suggest enhancing the paper writing to show more theoretical developments.
  5. Both linear regress and ANN are used to fit the experimental data. It would be better to provide their comparisons. Also, an ANN is a highly nonlinear model, but the results show that the relationship of both sets of data are quite linear. Thus, is it necessary to use an ANN?
  6. I believe that both linear regression and ANNs will be used to predict disturbances in applications. Thus, it would be better to test the fitted models by using another set of experimental data, which are totally different from the training data.

Author Response

Response to Reviewer 2 Comments

[Song Kang

Beihang University,

XueYuan Road No.37,

HaiDian District, BeiJing, China

kangsong@buaa.edu.cn]

Re: Response for manuscript applsci-1628455 “Estimation system of disturbance force and torque for underwater robot based on artificial lateral line” by Kang Song, Wusheng Chou, and Junhao Yu

Dear Reviewers :

Thank you very much for your time involved in reviewing the manuscript and your very encouraging comments on the merits.

Comments:

This paper presents an artificial lateral line system to estimate external disturbances of an underwater robot, where multiple linear regression and artificial neural network method are used to fit the mathematical models of the disturbances.

This study is interesting, and the authors use experimental data to support their results. Some minor things might be improved, which will enhance the qualities of this paper.

Comment 3:

The format and types of equations are not like those expressed in most literature. For instance, variables should be expressed in italic symbols, and symbol * is not usually used to express multiplication unless its refers to other specific mathematical operations. Or, maybe this journal defines a specific format and type for equations, and the authors have followed the rules.

Response 3:

Thanks for your reminder, we also noticed the format issue, and the template given by the journal misled us. The screen dump of the template is shown as follows. we have revised our format in this submission. Your advice helps a lot.

Comment 4:

It looks like that all equations provided are very fundamental, so this paper looks like that there are many theoretical theorems developed and provided. I would suggest enhancing the paper writing to show more theoretical developments.

Response 4:

At present, most of the theoretical researches on artificial lateral line system are relatively basic, and there is no deep theory to prove and derive.Our theoretical derivation in this paper is to prove that the system can be seen as linear roughly, and to provide theoretical support for the analysis using multiple linear regression method.

Comment 5:

Both linear regress and ANN are used to fit the experimental data. It would be better to provide their comparisons. Also, an ANN is a highly nonlinear model, but the results show that the relationship of both sets of data are quite linear. Thus, is it necessary to use an ANN?

Response 5:

The graph in our paper is not an input-output graph, but an output-output graph. The vertical axis is the predicted disturbance output of the multiple linear regression or ANN system, and the horizontal axis is the actual disturbance output measured by the force/torque sensor (here we made a mistake: the force/torque sensor was written as the pressure sensor). The predicted value of the multiple linear regression or ANN system should better be equal to the true value measured by the force/torque sensor, that is, the data points fall on the y=x line. Our purpose in drawing this graph is to visualize the deviation of the predicted value from the true value. The further the data points deviate from the y=x line, the worse the prediction of the system. Therefore, it can be seen from the results that the data points of the ANN system are closer to y=x than the data points of the multiple linear regression, that is to say, the prediction deviation of the ANN system is smaller than that of the multiple linear regression system. ANN has better disturbance prediction performance.

In order to avoid further misunderstanding, we have strengthened and revised the relevant parts. Thanks for your great suggestion on improving the accessibility of our manuscript. Much appreciate.

Comment 6:

I believe that both linear regression and ANNs will be used to predict disturbances in applications. Thus, it would be better to test the fitted models by using another set of experimental data, which are totally different from the training data.

Response 6:

Thank you for your suggestion. We have done multiple sets of different experiments, and divided the final data into test set and training set according to the source of the data. So the training data and the test data are from different experiments.The graphs in our paper all come from the data of the test set, the data in the training set is used for training, and did not participate in the test. Maybe our description of the images in the text is not suitable, like "fitting" or "regression", misled you think that we use the training set data, we will modify these representations, your suggestions are very helpful.

We would like to take this opportunity to thank you for all your time involved and this great opportunity for us to improve the manuscript. We hope you will find this revised version satisfactory.

Sincerely,

Song Kang, Wusheng Chou, and Junhao Yu
